# Simulation and Performance Evaluation of Laser Heterodyne Spectrometer Based on CO$_2$ Absorption Cell

**Tengteng Xia** [1,2], **Jiqiao Liu** [1,2,3,4], **Zheng Liu** [1,2], **Fangxin Yue** [1,3], **Fu Yang** [5], **Xiaopeng Zhu** [1,2,3,*] **and Weibiao Chen** [2,3,4]

1   Laboratory of Space Laser Engineering, Shanghai Institute of Optics and Fine Mechanics, Chinese Academy of Sciences, Shanghai 201800, China
2   Center of Materials Science and Optoelectronics Engineering, University of Chinese Academy of Sciences, Beijing 100049, China
3   Key Laboratory of Space Laser Communication and Detection Technology, Shanghai Institute of Optics and Fine Mechanics, Chinese Academy of Sciences, Shanghai 201800, China
4   Pilot National Laboratory for Marine Science and Technology (Qingdao), Qingdao 266237, China
5   College of Science, Donghua University, Shanghai 201620, China
*   Correspondence: xp_zhu@siom.ac.cn

**Abstract:** The laser heterodyne radiometer (LHR) has the advantages of miniaturization, low cost, and high spectral-resolution as a ground-verification instrument for satellite observation of atmospheric trace-gas concentration. To verify the accuracy of LHR measurements, a new performance evaluation method is presented here, based on an ASE source and a CO$_2$ absorption cell in the laboratory. Preliminary simulation analysis based on the system parameters of LHR is carried out for the performance analysis and data processing of this new combined test system. According to the simulation results, at wavelength deviation of fewer than 30 MHz, the retrieval error, which increases with bandwidth, can obtain an accuracy of 1 ppm within the bandwidth range of the photodetector (1.2 GHz) when this instrument line shape (ILS) is calibrated. Meanwhile, when the filter bandwidth is less than 200 MHz, the maximum error without ILS correction does not exceed 0.07 ppm. Moreover, with an ideal 60 MHz bandpass filter without ILS correction, LHR's signal-to-noise ratio (*SNR*) should be greater than 20 to achieve retrieval results of less than 1 ppm. When the *SNR* is 100, the retrieval error is 0.206 and 0.265 ppm, corresponding to whether the system uncertainties (temperature and pressure) are considered. Considering all the error terms, the retrieval error (geometrically added) is 0.528 ppm at a spectral resolution of 0.004 cm$^{-1}$, which meets the measurement accuracy requirement of 1 ppm. In the experiment, the retrieval and analysis of the heterodyne signals are performed for different *XCO$_2$* with [400 ppm, 420 ppm] in the absorption cell. Experimental results match well with the simulation, and confirm the accuracy of LHR with an error of less than 1 ppm with an *SNR* of 100. The LHR will be used to measure atmospheric-CO$_2$ column concentrations in the future, and could be effective validation instruments on the ground for spaceborne CO$_2$-sounding sensors.

**Keywords:** LHR; CO$_2$ absorption cell; ILS; *SNR*; *XCO$_2$*

## 1. Introduction

The increase in greenhouse gas (GHG) concentrations is the primary cause of global warming in the atmosphere. Global warming accelerates the melting glaciers and causes extreme weather disasters [1]. CO$_2$ is not only one of the primary GHGs but also a significant contributor to the global carbon cycle and radiation budget [2]. Therefore, long-term and accurate observation of CO$_2$ and other greenhouse gases is of great importance for developing appropriate mitigation plans and studying climate change. The observation mode of GHGs can be classified into satellite-based, airborne-based, and ground-based. Spaceborne measuring instruments, such as GOSAT and OCO-2, can provide CO$_2$ column concentrations (*XCO$_2$*) distribution information on a global scale [3–6]. However, the

temporal and spatial resolution is low, and they cannot achieve long-term observation in the local area. On the other hand, ground-based measurement methods are still indispensable, especially as a verification tool for spaceborne instruments [7]. Among them, the Fourier-transform spectrometer (FTS) is most commonly used because of its high spectral-resolution and measurement accuracy [8–10]. However, its large size and high-cost limit its application. The laser heterodyne radiometer (LHR), therefore, due to its miniaturization, low cost, and high spectral-resolution, is a suitable alternative as a portable measuring instrument.

The laser heterodyne radiometer (LHR) has been developed in recent years. The LHR system constructed by Weidmann et al., STFC Rutherford Appleton Laboratory, is mainly operated in the mid-infrared band [11–14]. The $O_3$ profile was retrieved using the ground-based prototype quantum-cascade laser LHR [11]. Moreover, an ultra-high-resolution (0.002 cm$^{-1}$) LHR system based on the external-cavity quantum-cascade laser (EC-QCL) has been developed for the detection of a variety of gas molecules ($H_2O$, $O_3$, $N_2O$, $CH_4$, $CCl_2F_2$) in the atmosphere [12,13]. Hollow waveguide technology was applied in the LHR [14] and spectral-channel optimization and early-performance analysis for the Methane Isotopologues measurement by Solar Occultation (MISO) [15] was carried out. The NASA Goddard Space Flight Center's near-infrared LHR utilized fiber optics to greatly reduce the difficulty of coupling and miniaturizing the system. Wilson et al. developed a mini-LHR for near-infrared $CO_2$ and $CH_4$ in the atmospheric column [16,17], and carried out field measurements. Wang et al. studied a 3.53 μm room-temperature interband-cascade LHR, which can simultaneously observe $CO_2$ and $CH_4$ in ground-based solar-occultation mode [18]. They also developed a fiber near-infrared LHR to observe $CO_2$ and $CH_4$ [19]. The near-infrared LHR developed by Deng et al. can measure $CO_2$, $CH_4$, $H_2O$, and $O_2$ with high resolution (0.066 cm$^{-1}$) [20,21].

However, the accuracy of these LHR measurements was evaluated by comparing their results with those observed by spaceborne instruments (such as GOSAT) or ground-based observation systems (such as TCCON). Nevertheless, no truth value of the gas concentration in the atmosphere can be referred to as an evaluation criterion for LHR. Beyond this, it is also difficult to analyze the effect of each critical-system parameter on the retrieved results, quantitatively. In contrast, a true value can be set in advance in the laboratory, and the performance evaluation of the LHR system will be possible. At the same time, the influence of some important parameters can be analyzed through simulation and experiment validation, and the simulation can provide an important reference for atmospheric measurement and instrument parameter optimization.

In this paper, an ASE light source and a specially-designed $CO_2$ absorption cell are used to simulate the absorption line of $CO_2$. Referring to the integrated-path differential absorption (IPDA) [22], the pure $CO_2$ pressure in the absorption cell is equivalent to $XCO_2$ in the atmosphere. This takes advantage of the fact that the method of the integrated-path differential-absorption optical depth ($DAOD$) of $CO_2$ is the same in both states, for equivalence. The LHR system is evaluated by analyzing the difference between the retrieval $CO_2$ pressure and the actual value. The paper is arranged as follows. Section 1 is the introduction. Section 2 mainly introduces the principle of the LHR system, the calibration experiment design, and the retrieval algorithm. In Section 3, the influence of the system parameters and algorithm on the retrieval results are considered, through simulation. In addition, the retrieval error is statistically analyzed, especially the influence of the filter bandwidth and signal-to-noise ratio ($SNR$). The simulation work in this section can provide a reference for filter selection and performance evaluation in the experiment. In Section 4, an experiment system based on the simulation results is built. The $XCO_2$ is set in the range of 400 ppm to 420 ppm. Discussion and conclusions are presented in Sections 5 and 6, respectively.

## 2. Methods

The LHR utilizes a narrow-linewidth local-oscillator (LO) laser to mix with a broad-band signal light, which achieves frequency down-conversion from optical frequency to

radiofrequency (RF). In the point-by-point scanning mode of LO, RF signals within the system bandwidth around the LO wavelength at each point are retained for future signal processing. The signal light's spectral information can be reproduced by processing the RF signal. The sunlight transmitted through the atmosphere contains a large amount of information about the absorption of atmospheric molecules. Therefore, analyzing the spectrum signal of sunlight can obtain atmospheric molecules' concentration and vertical profile. The basic principle of LHR has been described in detail by Weidmann et al. [23], and is not the focus of this paper.

The LHR system presented in this paper is designed for measuring $CO_2$ column concentration in the atmosphere. It is an all-fiber system in the near-infrared band with a sweeping range of 1571.895–1572.145 nm, which covers the $CO_2$ R18 absorption line. We develop an experimental prototype employing an indoor $CO_2$-absorption cell to analyze the performance of the LHR system, precisely. This $CO_2$ absorption cell is specially designed to charge pure $CO_2$ gas to simulate the $CO_2$ *DAOD* of the total atmosphere layer on the spaceborne platform, and has a length of 15.213-m.

We use an L-band ASE source (Connet, Shanghai, China, VASS-L-B) and an absorption cell charged with pure $CO_2$ to simulate the absorption of $CO_2$ in the atmosphere, then obtain the spectrum information in the sweep-frequency range through the all-fiber LHR system. Table 1 gives basic information on the ASE source, whose output power can be adjusted. A diagram of the experimental setup is shown in Figure 1. Light from the ASE source passes through a reflective collimator (Thorlabs, Newton, NJ, USA, RC08APC-P01) and enters the $CO_2$ absorption cell. After the absorption of $CO_2$, it is collected by a single-mode optical fiber through another reflective collimator. The collected signal light is split into two parts. An InGaAs photodetector (Thorlabs, DET01CFC/M) receives one part to monitor the energy fluctuation of the signal light. The other part is intensity-modulated at 800 Hz by an acousto-optic modulator (AOM, Gooch & Housego, Ilminster, UK, FIBER-Q) for subsequent coherent-heterodyne detection. The extinction ratio of the AOM is 50 dB. A near-infrared distributed-feedback (DFB) laser emitting around 1.572 μm (FITEL, Carrollton, GA, USA, FRL15DCWD) functions as the LO laser. The wavelength of DFB can be tuned by adjusting its temperature and current. A signal generator generates a ramp voltage signal to control the injection current of the laser, to realize the frequency sweeping. At the same time, it generates a trigger signal to control the data acquisition card. The DFB laser's output is split into two beams by a fiber beam-splitter: one as an LO laser for subsequent coherent detection, the other as a reference beam for intensity and wavelength monitoring. A photodetector and a wavemeter monitor intensity and wavelength changes, respectively. The multiple adjustable attenuators (AA) used in the system are designed to adjust the light intensity to suit different *SNR* requirements. The LO laser and the AOM-modulated signal light are mixed through a fiber coupler and superimposed on a photodetector (Thorlabs, DET01CFC/M) with a bandwidth of 1.2 GHz. The DC-Block isolates the DC term in the beat signal generated by the photodetector, and the reserved RF signal contains the ASE components within the detector bandwidth around the wavelength of the LO-laser sweep point. The RF signal first passes through a two-stage RF amplifier. Then the amplified RF signal is filtered by a bandpass filter, and finally a square-law detector (Herotek, San Jose, CA, USA, DHM020BB) measures the RF signal. In particular, the double-side bandwidth of the bandpass filter reflects the spectral resolution of the system. The generated low-frequency voltage signal with the modulation frequency as the characteristic frequency is demodulated by the lock-in amplifier (Zurich Instruments, Zürich, Switzerland, MFLI). The demodulated signal, the monitoring signal of the ASE source, and that of the LO laser are acquired synchronously.

**Table 1.** Important parameters of ASE source.

|  | Minimum | Typical | Maximum | Unit |
|---|---|---|---|---|
| Output Power | 10 | - | 100 | mW |
| Wavelength Range | 1570 | - | 1602 | nm |
| Spectral width (FWHM) | - | 32 | - | nm |
| Flatness of spectrum | - | - | 3.5 | dB |

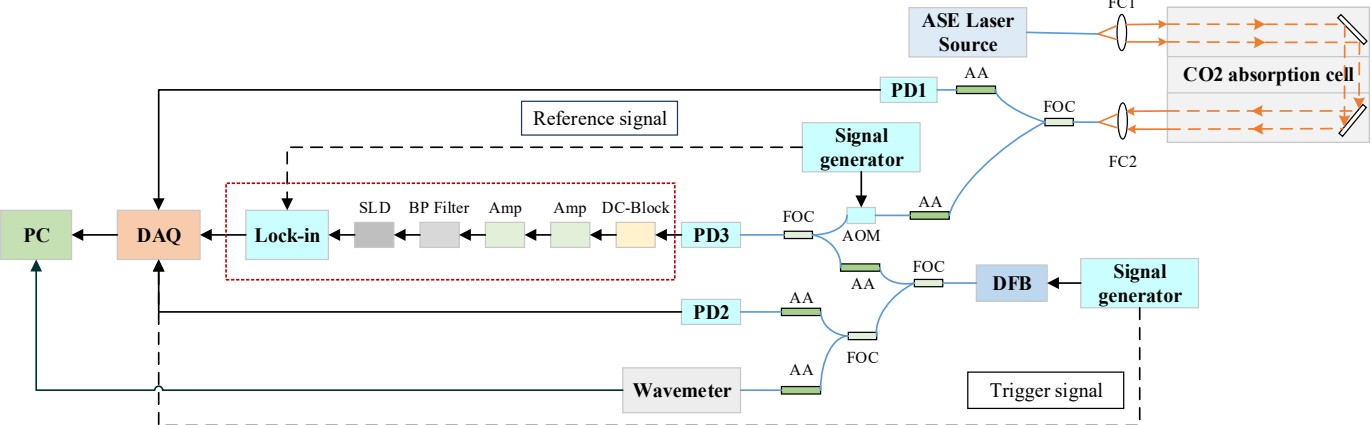

**Figure 1.** Diagram of experimental setup for LHR performance evaluation based on a $CO_2$ absorption cell. AA, adjustable attenuator; FOC, fiber-optic coupler; FC, fiber collimator, Amp, amplifier; BP Filter, bandpass filter; SLD, square-law detector; PD1, photodetector 1; PD2, photodetector 2; PD3, photodetector 3.

This specially-designed 15.213-m $CO_2$-absorption cell includes a $CO_2$-absorption-cell pipe, an optical- path turning structure, a temperature control system, a pressure detection system, a vacuuming system, and a $CO_2$ gas charging system [24]. The two 45° mirrors in the absorption-cell pipe give a total optical path of 15.213 m to the vertically incident beam. To evaluate the performance of LHR, it is necessary to analyze the correlation between the absorption-cell pressure and the $CO_2$ concentration in the atmosphere. For LHR, the absorption line in the absorption cell may have some differences from that in the real atmosphere. However, using the principle of path-integrated differential absorption (IPDA), the *DAOD* in the atmosphere can be equivalent to the *DAOD* of the absorption cell. The absorption cell can simulate the *DAOD* of atmospheric $CO_2$ concentration, due to its long optical path. The LHR may serve as a high-accuracy instrument on the ground to validate the performance of spaceborne-IPDA-lidar or passive-GHGs measurement instruments. The online and offline wavelengths of IPDA are selected in the strong- and weak-absorption regions of the $CO_2$ absorption line. The online and offline wavelengths are 1572.024 nm and 1572.085 nm, respectively, located on the R18 line [25]. Figure 2 shows the optical depth (OD) of $CO_2$ of the spaceborne platform and the corresponding absorption cell when the $CO_2$ column-averaged dry-air mixing ratio (*XCO$_2$*) is 400 ppm. Based on the principle of the space-borne IPDA lidar developed in our laboratory, the double-path *DAOD* and integrated weight function (*IWF*) for different concentrations are calculated [26]. The *DAOD* of the IPDA lidar and the absorption cell can be expressed as

$$DAOD = 2\int_{R_G}^{R_A} \rho_{CO_2}(r) \cdot \frac{P(r) \cdot N_A \cdot \Delta\sigma_{CO_2}(P(r), T(r))}{R \cdot T(r) \cdot (1 + \rho_{H_2O}(r))} dr \qquad (1)$$

$$DAOD_{CO_2} = \frac{P \cdot N_A}{R \cdot T} \cdot \Delta\sigma_{CO_2} \cdot L \qquad (2)$$

where $P$ is the pressure and $T$ is the temperature. $\Delta\sigma_{CO_2}$ is the differential absorption cross-section, which is related to the pressure and temperature distributions. $\rho_{CO_2}$ and $\rho_{H_2O}$

are the dry-air mixing ratio of $CO_2$ and $H_2O$, respectively. $N_A$ is Avogadro's number, and $R$ is the gas constant. $R_A$ and $R_G$ are the altitude of the satellite platform and the surface hard-target, respectively. In the absorption cell, the *DAOD* of the integrated path is directly converted to the product of the length of the absorption cell (*L*), due to the uniform temperature and pressure distribution. Combining *DAOD* with *IWF*, one obtains $XCO_2$ as

$$XCO_2 = \frac{DAOD}{2 \cdot IWF} \tag{3}$$

When the absorption cell is at a fixed temperature, the pure $CO_2$ charged with different pressures is the only variable that causes the change in *DAOD*. The US Standard Atmosphere model [27] and the spectroscopy database HITRAN 2020 [28] are used to simulate the *DAOD* of $XCO_2$ between 400 and 420 ppm, equivalent to that of the absorption cell under various pressures. The deviation of the charged pressure from the retrieved pressure can evaluate the accuracy of LHR by retrieving the measured heterodyne signals. Multiple sets of experiments with different $XCO_2$ can rule out chance.

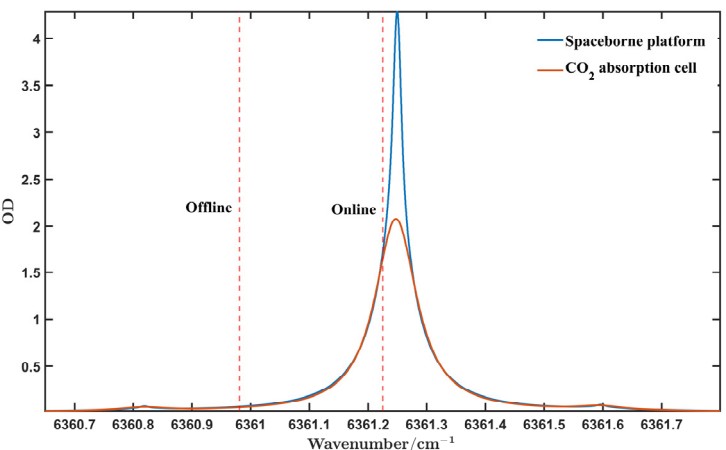

**Figure 2.** $CO_2$ optical depth of the spaceborne platform and the corresponding absorption cell.

The measured heterodyne signals are retrieved using the optimal-estimation method (OEM) [29]. In this experiment, the retrieved quantity is the pressure of the absorption cell, which is equivalent to $XCO_2$ in the atmosphere. The forward model, *F*, is described by

$$y = F(x, b) + \varepsilon \tag{4}$$

where *y* is the measurement vector, *x* is the state vector, and $\varepsilon$ is the error vector; *b* represents all other model parameters having an impact on the measurement. The forward model includes the transmission model of the absorption cell and the model of the LHR system, representing how the ASE optical signal containing the absorption information converts into the measurement signal and the associated noise. The OEM is based on the basic assumption of the multivariate-Gaussian-probability-distribution function. Because the problem is moderately nonlinear, combined with Bayesian statistics, the OEM is a Levenberg–Marquardt (LM) iterative algorithm based on the nonlinear-least-square method and minimizes the cost function to

$$\chi^2 = (y - F)^T S_\varepsilon^{-1}(y - F) + (x_i - x_a)^T S_a^{-1}(x_i - x_a) \tag{5}$$

where $S_\varepsilon$ is the measurement covariance matrix and $S_a$ is a priori covariance matrix; $x_a$ is a priori vector. The iterative formula of the state vector is

$$x_{i+1} = x_i + \left[ K_i^T S_\varepsilon^{-1} K_i + (1 + \gamma_i) \right]^{-1} \left[ K_i^T S_\varepsilon^{-1}(y - F(x_i)) - S_a^{-1}(x_i - x_a) \right] \tag{6}$$

where $K$ is the Jacobian matrix (or weighting functions), $\gamma$ is the Levenberg–Marquardt parameter, and the subscript represents the number of iterations.

## 3. Simulation Analysis

The absorption spectrum of $CO_2$ in the absorption cell is not the same as that of atmospheric $CO_2$, and the pressure in the absorption cell is uniform. The results of the simulation of LHR based on the absorption cell may have some limitations, but the idea of the simulation is the same. According to the calculation using the U.S. Standard Atmosphere model, when the $XCO_2$ is 400 ppm, the pressure of the $CO_2$ absorption cell is approximately 439 hPa. The following simulations are based on the premise of a 439-hPa charging pressure. According to the simulation, the influences of system bandwidth, wavelength shift, *SNR*, retrieval algorithm, and some systematic errors are studied. The statistical analysis of the errors is useful for subsequent experiments. Spectral resolution and *SNR* are two important parameters for characterizing LHR performance. The theoretical shot-noise-limited *SNR* of LHR can be expressed as [30,31]

$$SNR = \frac{2 \cdot \eta \cdot T_0 \cdot \sqrt{B \cdot \tau}}{\exp\left(\frac{h \cdot v}{k \cdot T_B}\right) - 1} \tag{7}$$

where $\eta$ is the effective quantum efficiency of the photodetector, and $T_0$ is transmission factor of the LHR system; $B$ is the system bandwidth, and $\tau$ is the integration time; $h$ is Planck's constant, and $k$ is Boltzmann's constant; $v$ is the frequency, and $T_B$ is the temperature of the black body. *SNR* is proportional to the square root of the bandwidth, but larger bandwidth means lower spectral resolution. Therefore, these two parameters, *SNR* and bandwidth, need to be balanced. In this paper, the *SNR* and spectral-resolution requirements are first considered separately and then combined to select a suitable filter.

### 3.1. Influence of Filter Bandwidth

In the point-by-point scanning mode of LO, the system bandwidth can be equivalent to the bandwidth of the bandpass filter, since the linewidth of LO can be negligible. The double-side bandwidth of the bandpass filter reflects the LHR system' spectral resolution. The spectral resolution of the system is kept constant during the LO frequency scan, and each sweep point can be controlled independently. The signal at each sweep point can be regarded as the integrated quantity of the spectral signal within the system's bandwidth near the LO wavelength. The system determines the range of integration, and therefore defines the instrument line shape (ILS). The ILS, a significant parameter in the forward model, mainly reflects the broadening effect of the system caused by bandwidth [32]. The measured heterodyne signal reflects the convolution of the actual spectral signal with the ILS of the LHR system. Figure 3 shows a schematic of the measurement process, visualizing the effect of the broadening effect of ILS. In principle, the ILS needs to be measured accurately. In addition, it is required to deconvolute the heterodyne signal before retrieving it. The smaller the bandwidth, the higher the spectral resolution, but the corresponding *SNR* will decrease. Therefore, selecting an appropriate filter bandwidth in the measurement process is necessary. Before analyzing the effects of other factors, the retrieval result is simulated for the ideal state of noiseless and infinitely small bandwidth. The retrieval error is only 0.001 ppm, which proves that the retrieval algorithm itself can achieve high accuracy.

However, when the bandwidth increases, the retrieval errors increase. The influences of different filter-bandwidths are analyzed in the retrieval results within the 1.2 GHz bandwidth of the photodetector, and the necessity of ILS correction is proposed. Figure 4 shows the influence of different filter-bandwidths on retrieval results, and compares the correction degree of the ILS correction. The extent of ILS correction is illustrated by comparing the errors with and without the ILS correction for different filter-bandwidths. Figure 4a shows the pressure error at 439-hPa, and the corresponding $XCO_2$ error is shown

in Figure 4b. If the ILS correction is not implemented, the error is larger than 1 ppm when the bandwidth is larger than 780 MHz. The retrieval result is significantly improved with ILS correction, and the error within the whole bandwidth of the detector (1.2 GHz) is less than 1 ppm. However, the smaller the bandwidth, the more accurate the retrieval results. When the filter is within 200 MHz, the maximum error with/without ILS correction is 0.006/0.069 ppm, respectively. Therefore, when the bandwidth is less than 200 MHz, the maximum error without ILS correction is less than 0.07 ppm. However, the actual bandwidth of the filter does not exactly match the nominal bandwidth. The ILS needs to be measured. The smaller the bandwidth, the more difficult it is to measure the ILS accurately. Therefore, the smaller the effect of ILS correction, the better.

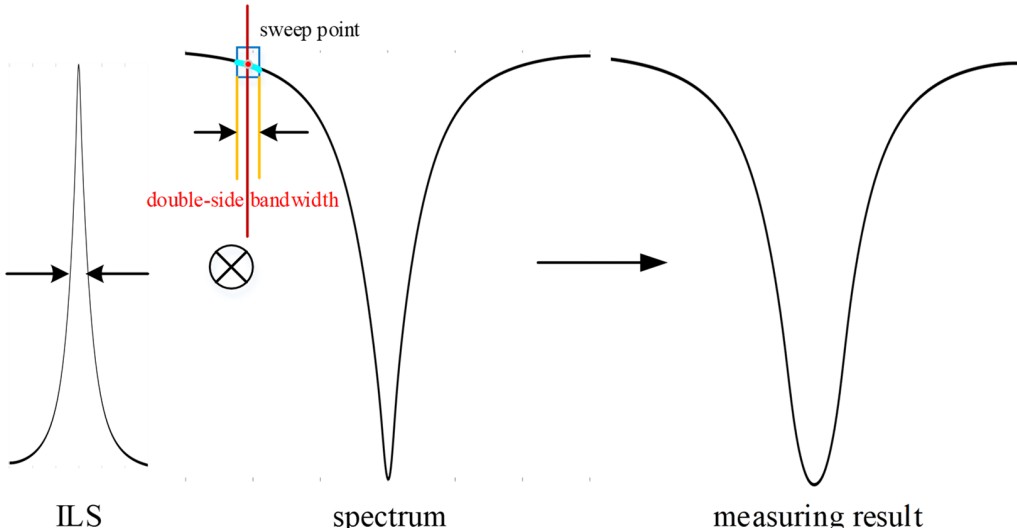

**Figure 3.** Schematic diagram of the measurement process.

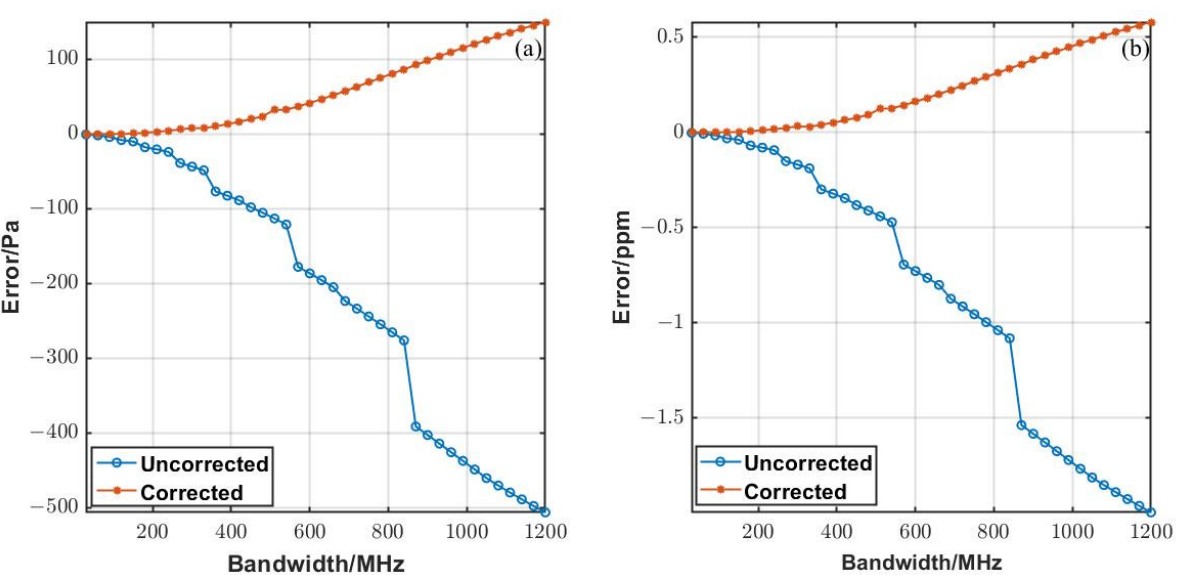

**Figure 4.** (**a**) Pressure error at different filter-bandwidths. (**b**) Equivalent $XCO_2$ error at different filter-bandwidths.

### 3.2. Influence of Wavelength Shift

Wavelength calibration of the original heterodyne signal has always been an important step. Wavelength calibration is generally performed using the absorption peak of the simulated spectrum and the measured heterodyne signal. Due to the limitations of the system and the correction algorithm, there may be some errors in wavelength calibration.

The effect of wavelength shift based on a minimum sweep-step of 30 MHz is analyzed. Figure 5a shows the error of the retrieval results with and without ILS correction, while Figure 5b shows that of the equivalent $XCO_2$.

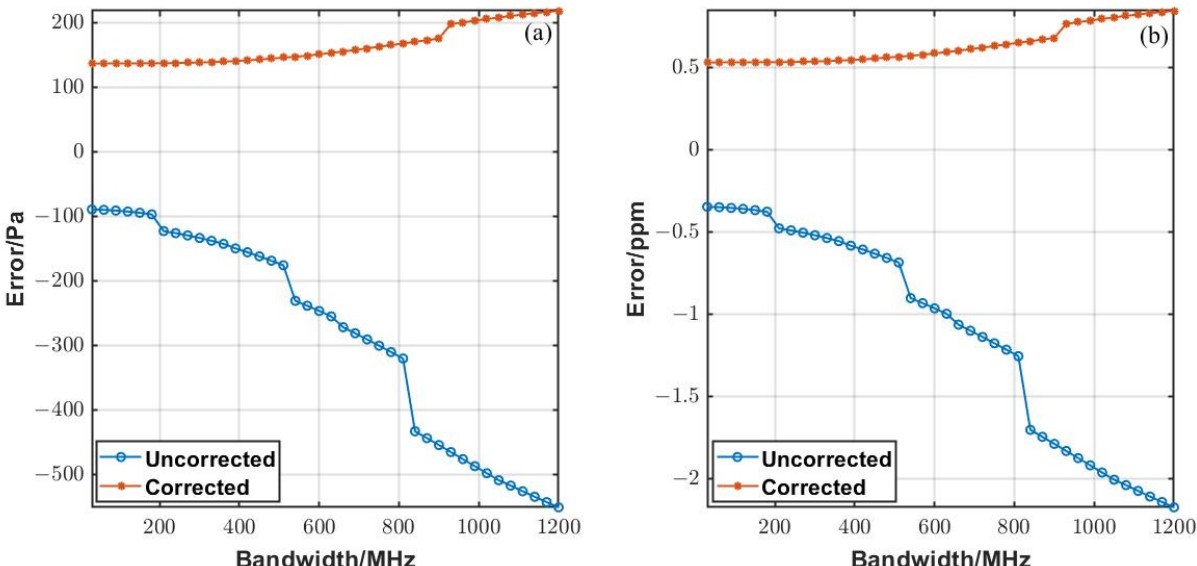

**Figure 5.** (**a**) Influence of wavelength shift on pressure error at different bandwidths. (**b**) Influence of wavelength shift on equivalent $XCO_2$ error at different bandwidths.

When the wavelength shift is 30 MHz, there is an error in the retrieval result which is uncorrected. The error without ILS correction is even smaller than that with ILS correction under low bandwidths. However, the error of retrieval results increases faster without ILS correction with the increase of bandwidths. The error is larger than 1 ppm when the bandwidth is larger than 600 MHz. The retrieval error with ILS correction can be within 1 ppm. When considering the effect of the wavelength deviation, the filter selection is within 200 MHz. In this case, the lack of ILS correction can compensate for some effects of wavelength deviation. When the filter bandwidth is within 200 MHz and the wavelength deviation is 30 MHz, the maximum absolute error is 0.533/0.375 ppm with/without ILS correction, respectively. For a bandwidth of 200 MHz, the corresponding double-side spectral resolution is ~0.013 cm$^{-1}$. Therefore, the bandwidth of the filter should be better within 200 MHz. The smaller the bandwidth of the bandpass filter, the smaller the error, if the $SNR$ can meet the requirements. These analyses provide a reference for the selection of filters in subsequent experiments.

### 3.3. Influence of SNR

According to the theoretical calculation and measurement, when the system is used for atmospheric $CO_2$ measurement, the $SNR$ remains greater than 100 for bandwidths greater than or equal to 30 MHz. As the actual filter-bandwidth may be greater than the nominal bandwidth, 60 MHz bandwidth provides some leeway. The ideal 60 MHz bandpass-filter is chosen as the simulation basis for the next simulation, where the error without ILS correction is only 0.008 ppm. Since the calculation method of $SNR$ in the actual measurement is slightly different, the ratio of spectral-absorption depth to the standard deviation of the baseline is taken as the $SNR$ here. The main purpose of simulating the influence of different SNRs is to find the boundary value of $SNR$ for good retrieval results. The influence of random Gaussian noise added to the forward model at different SNRs on the retrieval results is analyzed. Due to the randomness of noise, the simulation results can only be reference values, and cannot represent the absolute correlation between the $SNR$ and the retrieval bias. Fifteen sets of random heterodyne-signals are generated under each group of $SNR$, and the retrieval results are shown in Figure 6. The errors between

the simulated true concentrations (red line) and the average of the retrieval results (green dotted line) are shown separately for different SNRs. The errors and standard deviations are calculated under different SNRs, as shown in Table 2. The *SNR* should be greater than 20 to keep the multiply averaged errors less than 1 ppm. Figure 7 compares the differences between heterodyne signals from multiple simulations, with SNRs 20 and 60 as examples. The larger the *SNR*, the less the heterodyne signals deviate.

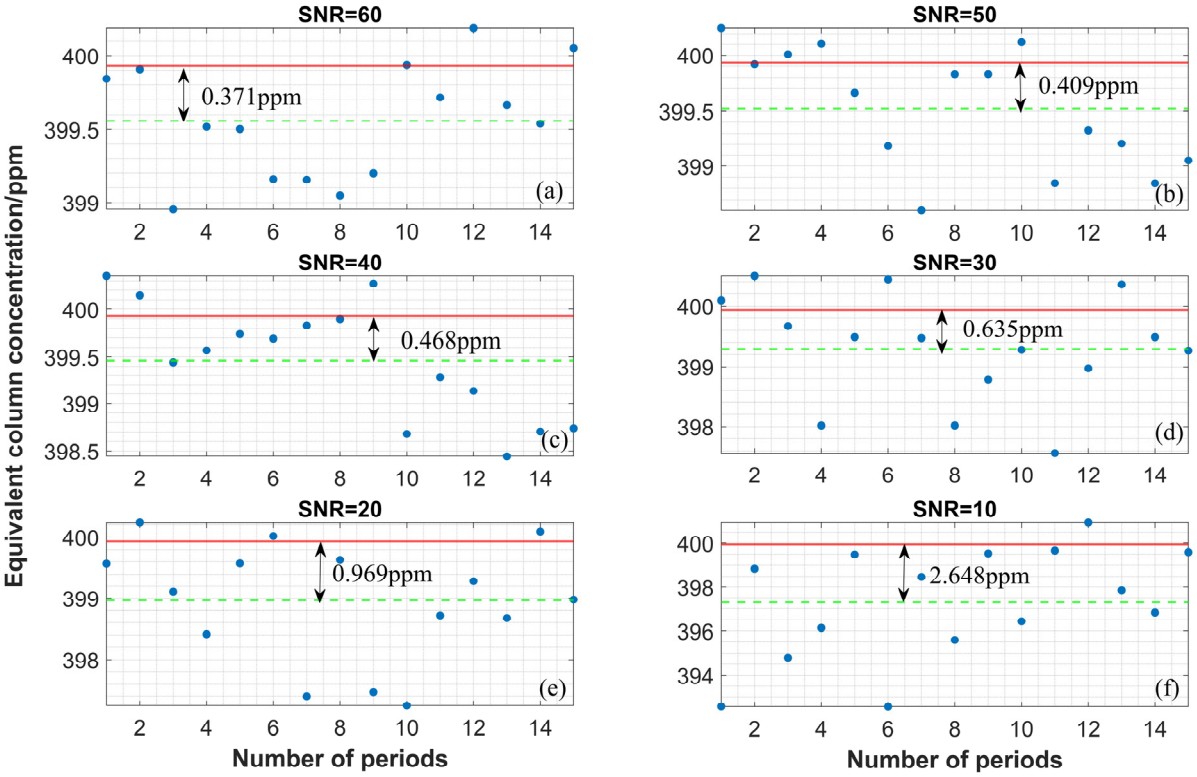

**Figure 6.** Retrieval results using *SNR* of (**a**) 60; (**b**) 50; (**c**) 40; (**d**) 30; (**e**) 20; (**f**) 10, respectively. The actual retrieval results (blue dot), averages of retrieval results (green dotted-line) and true values (red line) are shown.

**Table 2.** Retrieval results of different *SNR* statistics.

| SNR | Mean/ppm | Std/ppm | Error/ppm |
|---|---|---|---|
| 60 | 399.558 | 0.410 | −0.371 |
| 50 | 399.520 | 0.539 | −0.409 |
| 40 | 399.461 | 0.551 | −0.468 |
| 30 | 399.294 | 0.902 | −0.635 |
| 20 | 398.960 | 0.980 | −0.969 |
| 10 | 397.281 | 2.600 | −2.648 |

The double-side spectral resolution is approximately 0.004 cm$^{-1}$, corresponding to the bandwidth of 60 MHz. The output-power regulation of the ASE source ensures that the *SNR* of the LHR system is maintained at around 100. Random noise with Gaussian distribution is added, to simulate the spectra of multiple measurements at an *SNR* of 100. The retrieval results are analyzed and compared in Figure 8. The error and standard deviation caused by the retrieval analysis are 0.206 ppm and 0.198 ppm, respectively.

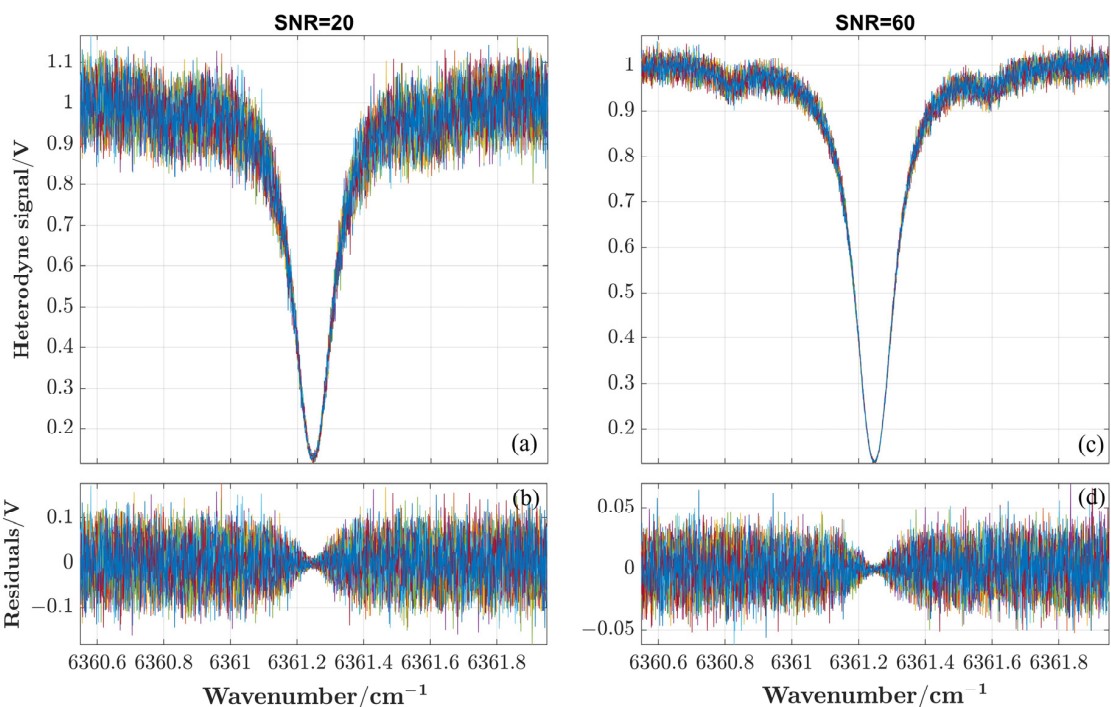

**Figure 7.** The simulated multiple heterodyne-signals at *SNR* of (**a**) 20, and (**c**) 60, respectively; the residuals between multiple heterodyne-signals at *SNR* of (**b**) 20, and (**d**) 60, respectively.

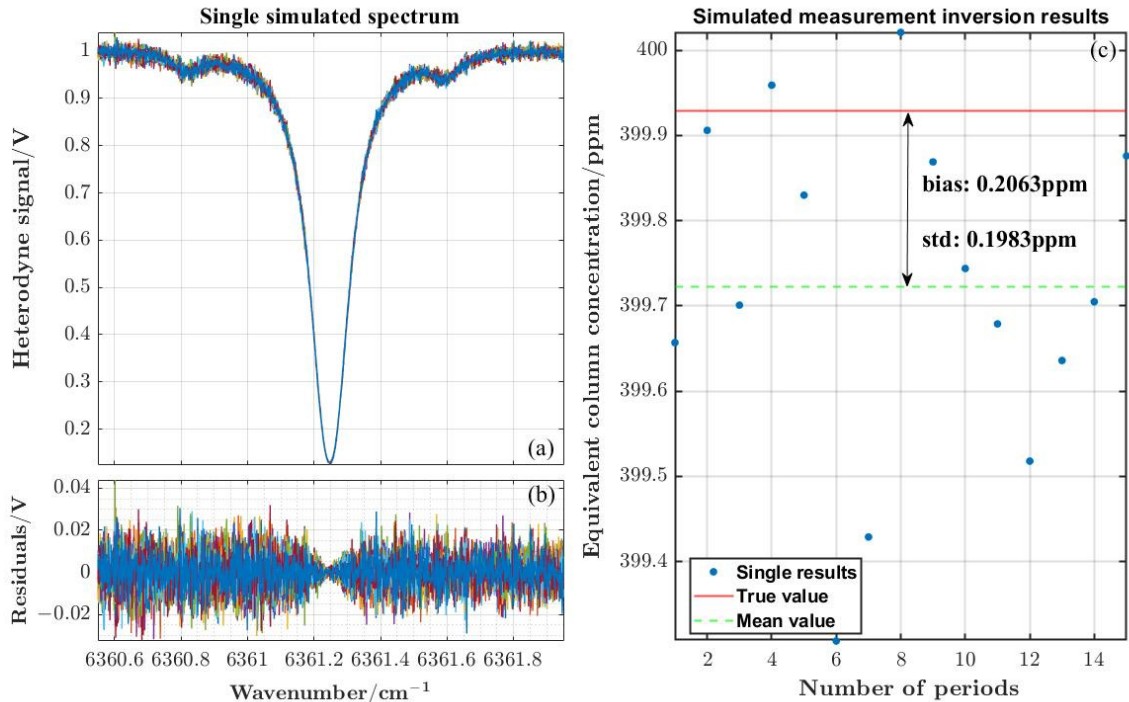

**Figure 8.** (**a**) The simulated multiple heterodyne-signals based on the *SNR* of 100. (**b**) The residuals between multiple heterodyne-signals. (**c**) Retrieval results of multiple heterodyne-signals.

### 3.4. Influence of LO Fluctuation

Based on the analyses in the previous section, ILS correction can be ignored at a spectral resolution of 0.004 cm$^{-1}$. Table 3 shows the critical-system parameters used in the simulation. The system is modeled, and the error caused by the system model and the retrieval algorithm together is 0.052 ppm when noise is not considered.

During the sweeping process, the unstable LO's power also causes errors in the retrieval results. Through experimental monitoring, the power-variation range is 12.5%, and the power instability of the LO is 2‰, within the frequency-sweep range. The LO power fluctuation is added to the forward model to analyze the error. After removing the error of the retrieval algorithm, the error is approximately 0.05 pp, due to the influence of the LO power fluctuation. Table 4 shows the error for different conditions.

**Table 3.** System parameter in the simulation.

| Category | Parameter | Value | Unit |
|---|---|---|---|
| DFB | Wavelength-sweep range | 1571.895~1572.145 | nm |
| | Average power | 1 | mW |
| Photodetector | Bandwidth | 1.2 | GHz |
| | Response | 0.95 | A/W |
| DC-Block | Bandwidth | 0.1–8000 | MHz |
| Amplifier | Gain | $13 \times 2$ | dB |
| Bandpass filter | Bandwidth | 60 | MHz |
| Square-law detector | Frequency range | 0.1–2000 | MHz |
| Lock-in amplifier | Reference frequency | 800 | Hz |
| | Integration time | 10 | ms |

**Table 4.** Influence of LO fluctuations on retrieval results.

| | $XCO_2$/ppm | Error/ppm |
|---|---|---|
| True value | 399.929 | |
| Ideal state | 399.980 | 0.052 |
| LO fluctuation | 400.030 | 0.101 |

At the same time, the effect of the LO power fluctuation on the *SNR* of 100 is analyzed. Considering the effect of LO-power-fluctuation noise, Figure 9 compares the retrieval results at the *SNR* of 100. The average retrieval error of multiple models is approximately 0.151 ppm.

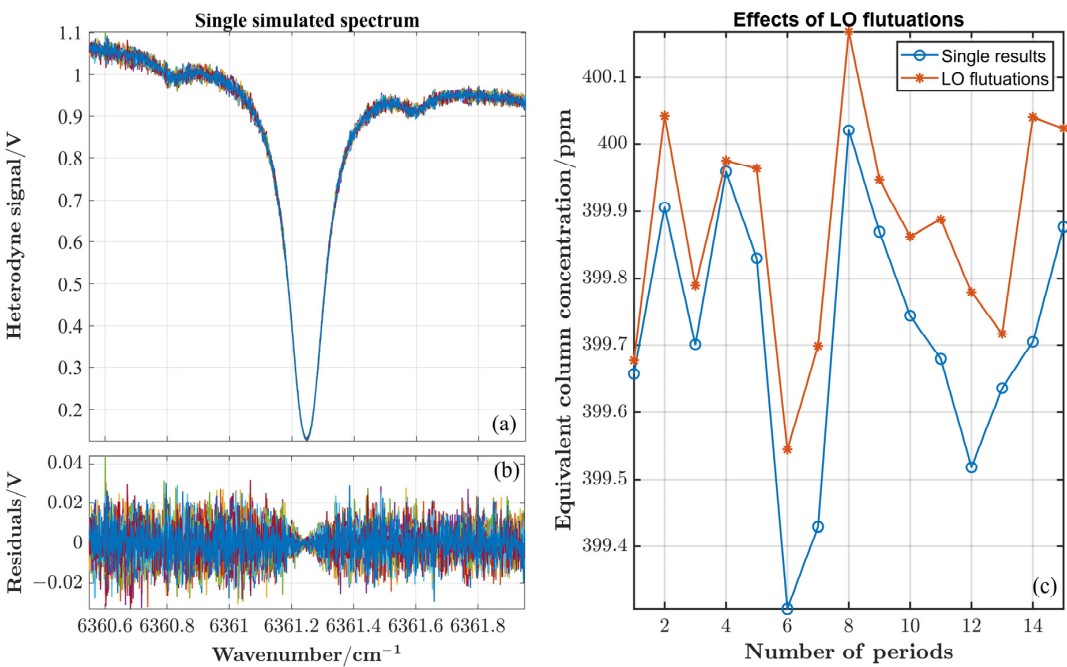

**Figure 9.** (**a**) The simulated multiple heterodyne-signals, based on the actual *SNR* and the LO fluctuation. (**b**) The residuals between multiple heterodyne-signals. (**c**) Comparison of the influence of the LO fluctuation on the results under the *SNR* of 100.

### 3.5. Influence of Temperature and Pressure Uncertainty

In the experimental setup, the absorption cell itself has some uncertainties, the influence of which it is necessary to evaluate. The absorption cell can then be judged as to whether it can be standard equipment for the LHR calibration in the laboratory. Table 5 shows some uncertainties in the absorption-cell temperature and pressure, parameters inherent to the construction of the absorption cell. In addition, these parameters are analyzed in this section.

**Table 5.** Index parameters of the absorption cell.

| Parameter | Precision | Value |
|---|---|---|
| Pressure measurement | 100 | Pa |
| Pressure control | 200 | Pa |
| Temperature control | 2 | K |
| Temperature measurement | 0.1~0.15 | K |

Given that the temperature measurement uncertainty is 0.1 K, the influence on the retrieval results is analyzed with an *SNR* of 100. In the forward, model regardless of noise, the error caused by temperature uncertainty is 0.04 ppm. In the *SNR* of 100, the error caused by temperature uncertainty under multiple averages is 0.07 ppm. The pressure measurement uncertainty of the absorption cell is 100 Pa, resulting in an error of 0.256 ppm. The error caused by the $CO_2$ absorption cell (temperature and pressure) is 0.265 ppm (geometrically added), which can meet the standard equipment's requirement. Errors caused by all uncertainties are statistically analyzed, and the results are shown in Table 6. The geometric sum of all error terms is 0.528 ppm, which is of great help in the subsequent experimental work. The simulation results have some limitations, but the analysis method can be used for column-concentration measurements of atmospheric $CO_2$.

**Table 6.** Contributions of various error terms of LHR verification.

| Error Term | Error/ppm | Uncertainty for $XCO_2$ |
|---|---|---|
| Retrieval algorithm | 0.052 | - |
| Bandwidth | 0.006 | 60 MHz |
| Wavelength shift | 0.375 | 30 MHz |
| *SNR* | 0.206 | 100 |
| LO fluctuation | 0.151 | 2‰ |
| Temperature | 0.07 | 0.1 K |
| Pressure | 0.256 | 100 Pa |
| Error budget (geometrically added) | 0.528 | - |

## 4. Experimental Results and Discussion

### 4.1. Experimental Setup and SNR Analysis

To validate the performance of the LHR, five groups of experiments are implemented at a range of 400 ppm to 420 ppm. Figure 10 shows a diagram of the experimental setup, which includes the principle prototype of the LHR and the $CO_2$ absorption cell. Table 7 lists both the theoretical pressure and actual pressure in the absorption cell when $XCO_2$ is changed from 400 to 420 ppm.

The RF signals of beat-frequency signal, LO, signal-light, and no-light state are measured. The RF signal's frequency spectrum, as shown in Figure 11, is measured with a spectrum analyzer. The main noise-frequency-range is between 300 MHz and 800 MHz, so the passband range of the bandpass filter should be less than 300 MHz. In the LHR system, DC-140 MHz and 58–82 MHz filters are selected, due to the lack of a suitable filter with a 30 MHz bandwidth. This combination can avoid the upward warps of the 58–82 MHz filter in the high-frequency part of the transmittance curve. The ideal 58–82 MHz bandpass filter

should correspond to a double-side-bandwidth spectral resolution of ~0.0016 cm$^{-1}$, and ILS correction can be ignored during retrieval.

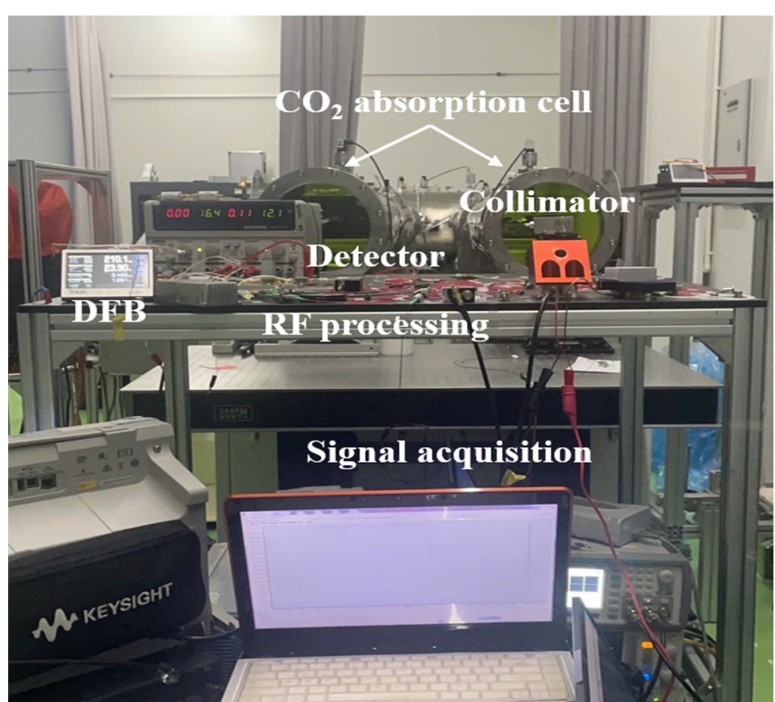

**Figure 10.** Diagram of experimental device.

**Table 7.** The pressure in the absorption cell and the corresponding $XCO_2$.

| Theoretical Value | | Actual Value | |
|---|---|---|---|
| $XCO_2$/ppm | Theoretical pressure/hPa | $XCO_2$/ppm | Actual pressure/hPa |
| 400 | 439.2 | 399.812 | 438.7 |
| 405 | 452.6 | 405.186 | 453.1 |
| 410 | 467.1 | 409.961 | 467.0 |
| 415 | 483.0 | 414.993 | 483.0 |
| 420 | 500.6 | 420.103 | 501.0 |

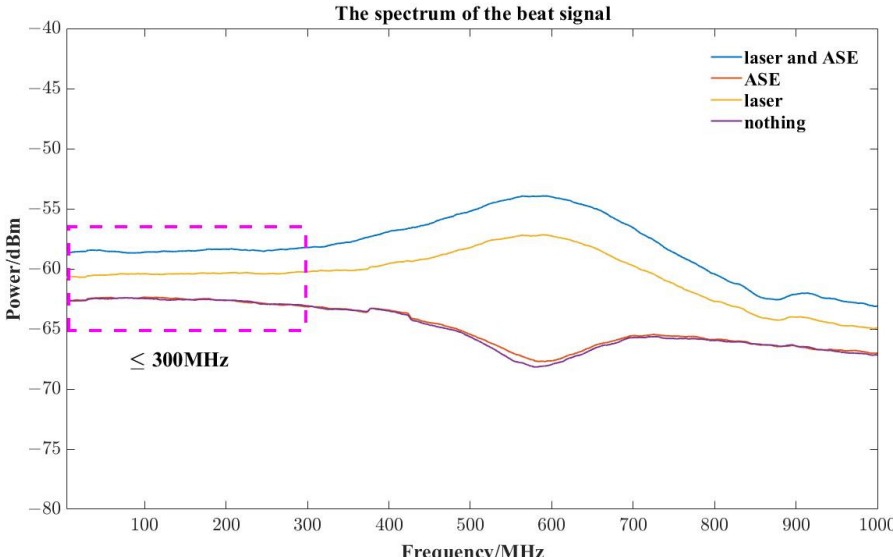

**Figure 11.** The spectrum of the beat signal.

Each group is measured at different pressures, 15 times, in the experiment. The *SNR* of measured heterodyne signals is analyzed with a pressure of 438.7 kPa. Figure 12a shows the heterodyne signals measured at a pressure of 438.7 kPa, and the residuals between all measurement periods are shown in Figure 12b. The *SNR* of the experimental LHR system is approximately 100.

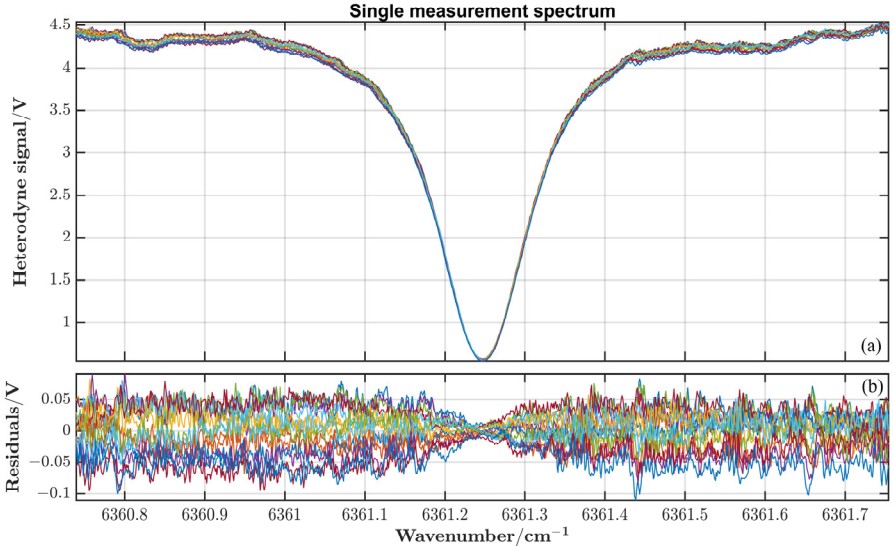

**Figure 12.** (**a**) The heterodyne signals of 15 times at 438.7 hPa. (**b**) The residuals between multiple measurement-signals.

### 4.2. Results

The original signal needs to be preprocessed. It mainly consists of three parts: correction of LO, background DC noise, and wavelength shift. ASE source monitoring is used for distinguishing the effectiveness of the heterodyne signals. The collected signals are removed when the ASE source fluctuation exceeds 5%. In the actual experimental process, the maximum uncertainty of the ASE source is less than 2%, which meets the accuracy requirements. The background noise is converted into a certain DC bias, and superimposed on the heterodyne signals. In the actual test process, each sweep cycle takes 2 s to collect the heterodyne signal of no LO, which can be used as the background noise. The DC bias needs to be corrected before retrieving the heterodyne signals. Otherwise, it will cause a large bias in the retrieval result. From the previous analysis of wavelength deviation, wavelength calibration affects the retrieval results tremendously. The wavelength alignment should be as accurate as possible. The wavelength calibration is carried out through maximum correlation, during data preprocessing.

The heterodyne signals of each group are preprocessed and compared with two different processing methods. One method is to retrieve the single heterodyne signal and analyze the standard deviation distribution. Another is to average the heterodyne signals of multiple periods, to improve accuracy. Figure 13 shows the distribution of retrieval results of five sets of single measured signals and their errors, from the truth values. The results are summarized in Table 8. A linear fit is performed for the true and retrieval $XCO_2$ concentrations, and the correlation coefficient reaches 0.997, as in Figure 14. The root-mean-square error (RMSE) between the true and retrieval concentrations is only 0.54 ppm.

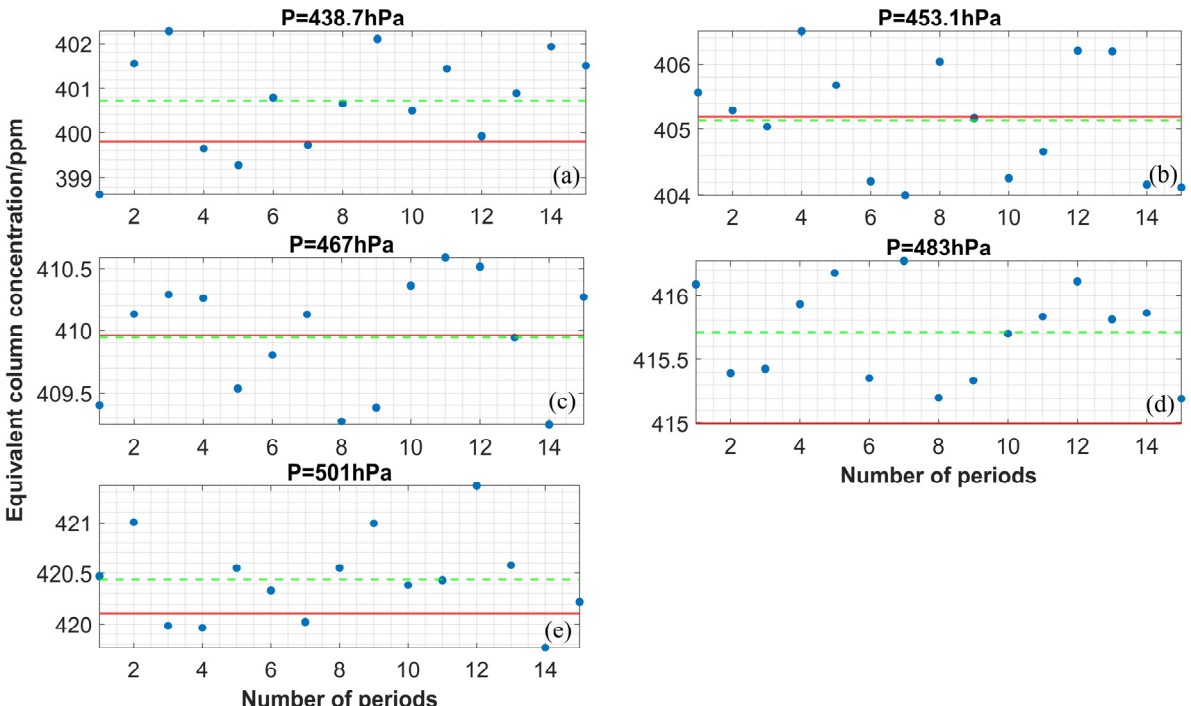

**Figure 13.** The retrieval results of 15 periods at (**a**) 438.7 hPa; (**b**) 453.1 hPa; (**c**) 467 hPa; (**d**) 483 hPa; (**e**) 501 hPa, respectively. The actual retrieval results (blue dot), averages of retrieval results (green dotted-line) and true values (red line) are shown.

**Table 8.** Comparison of retrieval results.

| | Method 1 | | | Method 2 | |
|---|---|---|---|---|---|
| Pressure/hPa | Mean/ppm | Std/ppm | Error/ppm | $XCO_2$/ppm | Error/ppm |
| 438.7 | 400.724 | 1.099 | 0.912 | 400.732 | 0.919 |
| 453.1 | 405.138 | 0.867 | −0.047 | 405.186 | −0.047 |
| 467.0 | 409.941 | 0.468 | −0.020 | 409.964 | 0.003 |
| 483.0 | 415.713 | 0.368 | 0.720 | 415.719 | 0.726 |
| 501.0 | 420.440 | 0.435 | 0.337 | 420.461 | 0.361 |

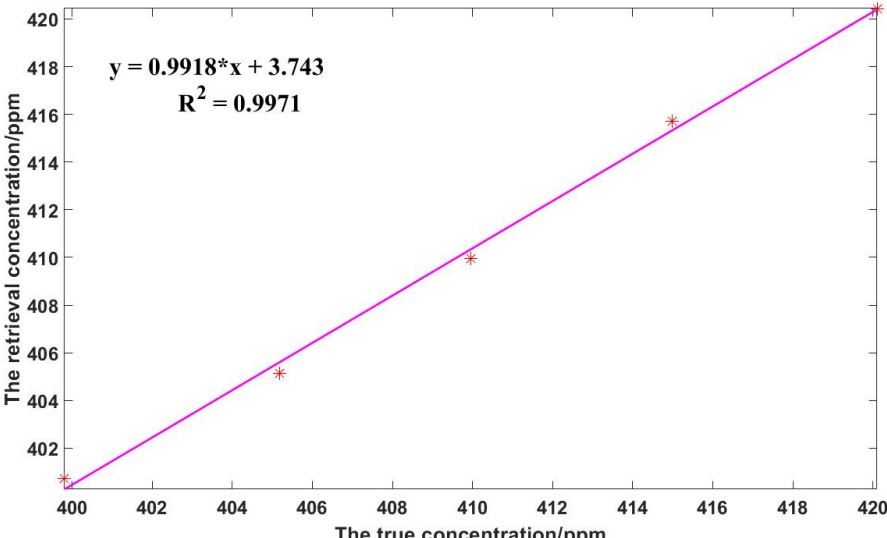

**Figure 14.** Linear fit of true and retrieval concentrations. The red * indicates the retrieval concentration corresponding to each true concentration.

The deviations between the averaged heterodyne-signal and the best-fit model curves are shown in Figure 15a–e. Figure 15f shows the differences in averagely measured heterodyne-signals under different pressures. The fitting curves have a close consistency with the heterodyne signals, and the maximum residuals are less than 2.5%. For the second processing method, the retrieval results are also listed in Table 8. The retrieval results of the two processing methods are close, and the heterodyne-signal averaging could obtain high accuracy. The LHR system can achieve 1 ppm accuracy, which verifies its performance with high accuracy. The LHR system can be used for future atmospheric $XCO_2$ measurements as an effective ground-verification device for satellite observations. Moreover, the experimental results are close to the simulation results, which verifies the model's validity.

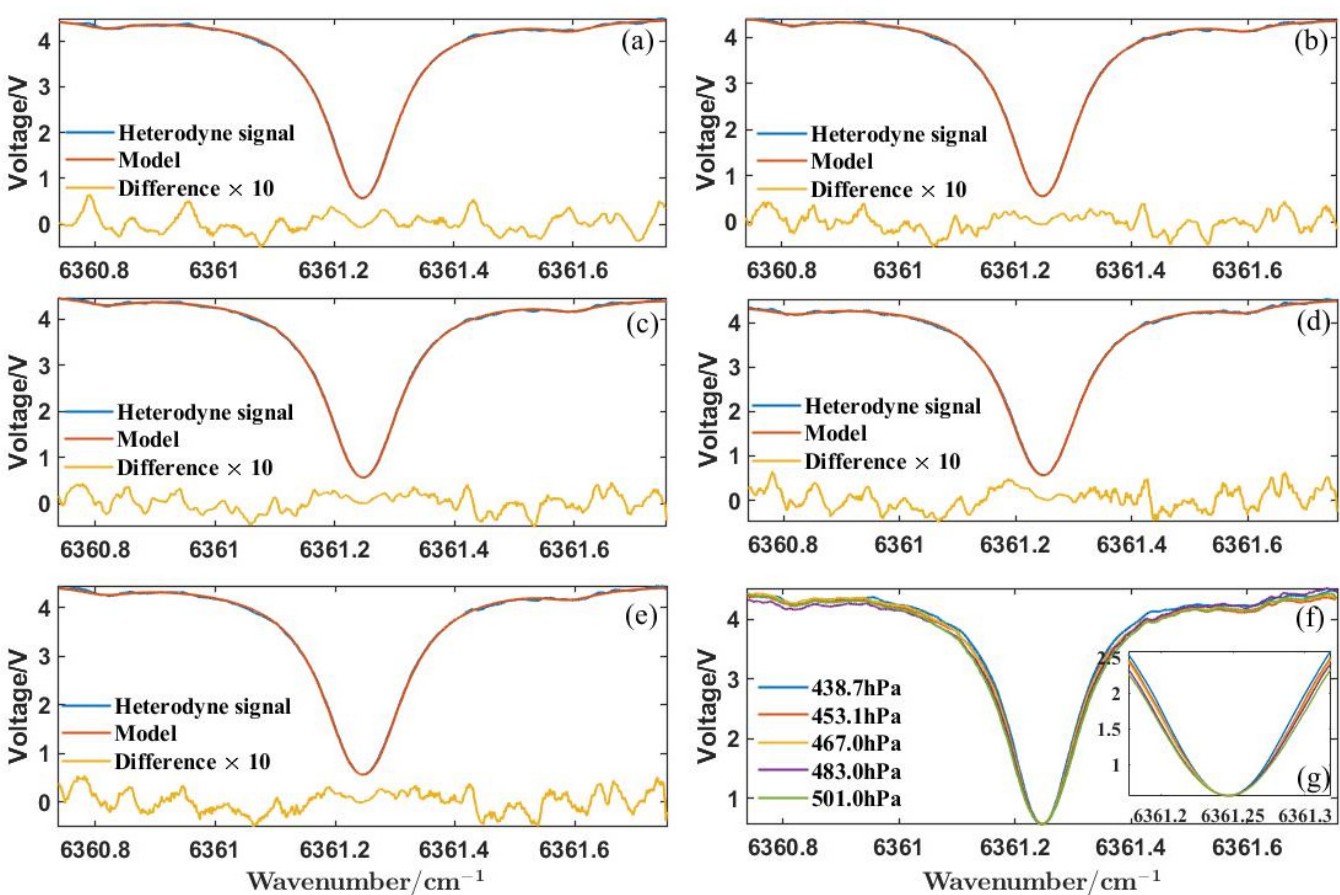

**Figure 15.** The experimental average heterodyne-signals and the model-fitting results at (**a**) 438.7 hPa, (**b**) 453.1 hPa, (**c**) 467.0 hPa, (**d**) 483.0 hPa, (**e**) 501.0 hPa. (**f**) The differences in the average measured-heterodyne-signals at different pressures. (**g**) The partially enlarged view of (**f**).

## 5. Discussion

In this paper, the LHR system is modeled, and the influence of important parameters is analyzed. The simulation analysis provides an important reference for appropriate system-parameter selection. The simulation method can guide atmospheric $CO_2$ measurement, but the current simulation results have some limitations, due to the inhomogeneity of the atmosphere. The simulation and experimental system based on the $CO_2$ absorption cell cannot fully simulate the absorption spectrum of $CO_2$ column concentration in the atmosphere. More relationships between bandwidth and *SNR* need to be established by the simulation analysis of atmospheric $CO_2$ measurements.

The *DAOD* of $CO_2$ in the absorption cell is equated to the *DAOD* in the atmosphere by the principle of IPDA. The pressure in the absorption cell could be adjusted, to change the *DAOD*. Experiments with different concentrations, from 400 ppm to 420 ppm, are implemented, and the measurement accuracy of the LHR system is well-proven with less

than 1 ppm. It is worth mentioning that *SNR* is controllable, because the signal light is simulated with an ASE source of adjustable power. The *SNR* of the system in the actual atmospheric observation is higher than 100, so a better performance is possible to realize.

## 6. Conclusions

The LHR has unique advantages over FTS. Although several research teams have built their LHR systems, they are not yet commercially available. The accuracy of the LHR measurement is mainly evaluated by comparison with other instruments. In addition, the extent to which important instrument-parameters affect the observations has not been quantified. A new performance-evaluation method is proposed, based on a $CO_2$ absorption cell here. In other cases, simulations are carried out to optimize the system parameters. The advantage of this method is that the true value is a criterion for evaluation. We have built an LHR system and have attempted to evaluate its performance before conducting atmospheric-observation experiments. At the same time, some important instrument parameters are quantified, and the error terms are analyzed and compared. Not only could these parameters be optimized, but the performance improvement method is also presented, for subsequent LHR field observations.

Simulation analysis is performed with the LHR system with a $CO_2$ absorption cell. The sensitivity analysis is performed using the actual LHR-system parameters. The filtering bandwidth affects the retrieval accuracy and the effectiveness of ILS correction, for which some analyses have been performed. When the filter bandwidth is 200 MHz, i.e., the spectral resolution is 0.013 cm$^{-1}$, the maximum retrieval error without ILS correction is 0.07 ppm. Selecting a bandpass filter with low bandwidth can simplify the ILS correction procedure. With an ideal 60 MHz bandpass-filter without ILS correction, LHR's *SNR* should be greater than 20 to meet the 1 ppm accuracy requirement. Based on the *SNR* of 100 and 60 MHz bandwidth, the error is ~0.206 ppm. The system's uncertainties regarding temperature and pressure cause a geometrically added error of 0.265 ppm. In a statistical analysis of the main error terms, the geometrically added error is 0.528 ppm, which can meet the accuracy of 1 ppm.

LHR performance is tested by simulating the change in $XCO_2$ from 400 to 420 ppm, corresponding to changing the pressure in the absorption cell. Then the heterodyne signals are retrieved. The error of the retrieval results is less than 1 ppm for different concentrations, and the high accuracy of the LHR is validated. The correlation between the true concentration and the retrieval concentration is as high as 0.997, and the RMSE is only 0.54 ppm.

In this paper, the calibration experiment based on the $CO_2$ absorption cell for the LHR is carried out to validate the measurement ability, which is useful for the subsequent measurements of the atmospheric solar-absorption-spectrum. In addition, the simulation based on two significant parameters (bandwidth and *SNR*) can provide an important reference for atmospheric measurement and instrument-parameters optimization. The actual experimental results verify the fact that the performance of the LHR system can meet the measurement requirements with high accuracy. The retrieval algorithm and correction method are helpful for future atmospheric $CO_2$ measurements.

**Author Contributions:** T.X. and J.L. proposed the experimental protocol; T.X. and Z.L. conducted the experiments; J.L. and X.Z. guided the experiments; T.X. performed the simulation analysis, processed the data results and wrote the manuscript; F.Y. (Fangxin Yue) and F.Y. (Fu Yang) modified the grammar and format of the whole article; J.L. and W.C. guided the writing of manuscripts. All authors have read and agreed to the published version of the manuscript.

**Funding:** This research was funded by the Strategic Priority Research Program of the Chinese Academy of Sciences, grant number XDA19090100; the International Partnership Program of Chinese Academy of Sciences, grant number 18123KYSB20210013; the Shanghai Science and Technology Innovation Action Plan, grant number 22dz208700; and the Shanghai Natural Science Foundation 22ZR1402600.

**Data Availability Statement:** Data used in this study are available from the corresponding author upon request.

**Acknowledgments:** The authors would like to thank the science teams of HITRAN for providing high quality and accessible data used in this study.

**Conflicts of Interest:** The authors declare no conflict of interest.

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
