# Peer review of "Simulation and Performance Evaluation of Laser Heterodyne Spectrometer Based on CO2 Absorption Cell"

_remotesensing, doi:10.3390/rs15030788_

Round 1

Reviewer 1 Report

Abstracts presents too many words. Hence recommending for reduction.

what is the meaning of heterodyne?

what kind of laser is used in LHR and its wavelength.

literature review is not well presented and hence in deep investigation should be conducted to describe what had done regional and global level

objectives should be rewritten as they are not clear to the readers.

methodology and simulation used in the present work is nicely presented.

in depth interpretation is required for few figures such as Fig 4, 6, and others.

Also, figures 7 and 8 looks similar with out discrimination adn hence should be presented with more details. however, i suggest the authors to relook in to the measurements and data for correction.

there are too many tables in teh paper and hence suggested to move few tables which are not important relevance to the supplementary information

figure 10 should be smoothed to correct the signal and SNR.

There are too many figures also and too much of redunancy.

At several instance, teh grammar should be corrected.

Lack of consistency with the results presented in figures adn tables and that interpreted in the text. kindly check the same

citation list should be verified and recommending to update teh citation list with the recent literature.

Reviewer 2 Report

1.Please supplement the requirements for technical indicators of the instrument and compare them with those of previous instruments.

2.Please supplement the physical photos of the instrument and the physical photos of the test site.

Reviewer 3 Report

The reviewer would like to thank the authors for this thoughtful manuscript. This work has good potential. The authors are requested to put in some additional efforts to improve the quality of this manuscript. 

Introduction 

The authors are requested to cite the following Earth observation articles reporting the influence of atmospheric warming and global energy balance on terrestrial ecosystems. Please elaborate on how the proposed methodology can assist such investigations and include some socio-economic significance for measuring CO2 concentration in the atmosphere on the lines of atmospheric warming and greenhouse effect.

-Shugar et al, A massive rock and ice avalanche caused the 2021 disaster at Chamoli, Indian Himalaya, Science, 2021.

-Tsai, Y.L.S., et al., 2019. Remote Sensing of Snow Cover Using Spaceborne SAR: A Review. Remote Sensing.

Evaluation Metrics

The authors are requested to compute performance evaluation metrics like RMSE, correlation coefficient, and MBE (bias), which are significant measures for such investigation. With regards to model evaluation, the authors are requested to discuss the effectiveness of performance metrics (like RMSE etc.) as demonstrated in the following interdisciplinary articles and discuss the use of these metrics for comparing modeled and observed data. 

i) Hastie et al., 2009. The elements of statistical learning: Data Mining, Inference, and Prediction

ii) Muhuri et al., 2021. Performance Assessment of Optical Satellite-Based Operational Snow Cover Monitoring Algorithms in Forested Landscapes, IEEE JSTARS.

iii) Valipour and Dietrich, 2022. Developing ensemble mean models of satellite remote sensing, climate reanalysis, and land surface models.

Figures

The authors are requested to provide figures with a higher resolution. The information presented in the figure is not very clear and the quality of some are rather fuzzy. Furthermore, the captions are not detailed enough for a complete understanding of the results presented in the figures. The authors are also requested to provide legends with clear symbology. 

Correlation

The authors indicate a correlation of 0.9971 in Fig. 13 with just three data points. Is this justified?

Conclusion 

The authors are requested to list the key contributions in this section. At the moment the section is not detailed enough. Furthermore, the authors are requested to separate Results and Discussion.
